

# Effects of maternal taurine supplementation on maternal dietary intake, plasma metabolites and fetal growth and development in cafeteria diet fed rats

Arzu Kabasakal Çetin, Tuğba Alkan Tuğ, Atila Güleç and Aslı Akyol

Department of Nutrition and Dietetics, Faculty of Health Sciences, Hacettepe University, Ankara, Türkiye

Corresponding author
Aslı Akyol,
asli.akyol@hacettepe.edu.tr

## ABSTRACT

**Background:** Maternal obesity may disrupt the developmental process of the fetus during gestation in rats. Recent evidence suggests that taurine can exert protective role against detrimental influence of obesogenic diets. This study aimed to examine the effect of maternal cafeteria diet and/or taurine supplementation on maternal dietary intake, plasma metabolites, fetal growth and development.

**Methods:** Female Wistar rats were fed a control diet (CON), CON supplemented with 1.5% taurine in drinking water (CONT), cafeteria diet (CAF) or CAF supplemented with taurine (CAFT) from weaning. After 8 weeks all animals were mated and maintained on the same diets during pregnancy and lactation.

**Results:** Dietary intakes were significantly different between the groups. Both CAF and CAFT fed dams consumed less water in comparison to CON and CONT dams. Taurine supplementation only increased plasma taurine concentrations in CONT group. Maternal plasma adiponectin concentrations increased in CAF and CAFT fed dams compared to CON and CONT fed dams and there was no effect of taurine. Hyperleptinemia was observed in CAF fed dams but not in CAFT fed dams. Malondialdehyde was significantly increased only in CAF fed dams. Litter size, sex ratio and birth weight were similar between the groups. There was an increase in neonatal mortality in CONT group.

**Discussion:** This study showed that maternal taurine supplementation exerted modest protective effects on cafeteria diet induced maternal obesity. The increased neonatal mortality in CONT neonates indicates possible detrimental effects of taurine supplementation in the setting of normal pregnancy. Therefore, future studies should investigate the optimal dose of taurine supplementation and long term potential effects on the offspring.

## INTRODUCTION

Obesity has become a significant public health problem due to its increasing prevalence at an alarming rate (*Ng et al., 2014*). Lifestyle changes, including increased energy intake and

decreased physical activity are considered as the main contributors of this outcome (*Popkin, 2015*). The rate of obesity and overweight in women of child-bearing age is also increasing (*Fisher et al., 2013*). Previous studies suggested that children born to mothers with maternal obesity are at higher risk for unfavorable birth outcomes and health problems (*Nagl et al., 2017*; *Ayonrinde et al., 2017*). The identification of obesity related complications during pregnancy and developing effective interventions as early as possible to prevent the development of childhood obesity is vital (*Blake-Lamb et al., 2016*). For this purpose, animal models of obesity are crucial to examine the mechanisms involved in the progression of obesity.

Through the developmental origins of health and disease paradigm, accumulating evidence suggest that maternal obesity or over nutrition during pregnancy results in development of components of metabolic syndrome in the offspring (*Desai et al., 2014*; *Raipuria et al., 2015*; *Dias-Rocha et al., 2018*). In particular, offspring born to dams fed a hyper-energetic cafeteria diet during gestation and/or lactation exhibited increased body weight (*Benkalfat et al., 2011*), reduced muscle force (*Bayol et al., 2009*), altered hepatic gene expression in insulin signaling pathway (*Daniel et al., 2014*) and different behavioral parameters (*Speight et al., 2017*). Studies reported that these effects could be reversed by specific interventions such as increasing maternal physical activity (*Son et al., 2019*) or including bioactive food components to obesogenic diets (*Sheen et al., 2018*).

Taurine, 2-aminoethane sulfonic acid, is involved in various metabolic pathways both in human and rodents (*Nielsen et al., 2017*). It serves crucial functions in biological processes that are related with detoxification, membrane integrity, bile acid conjugation, calcium levels and regulation of osmosis (*Ince et al., 2017*). Recently, few studies reported that taurine may exert a protective influence on oxidative stress induced by different disruptors in rodent (*Rashid, Das & Sil, 2013*; *Zheng et al., 2017*). More specifically, one study reported that offspring exposed to a maternal obesogenic diet with taurine supplementation during pregnancy and lactation had partially recovered pro-inflammatory hepatic profile (*Li et al., 2013*). Similar rescuing effect of taurine was also observed in maternal fructose-induced obesity (*Li et al., 2015*) and low protein diet models (*Larsen et al., 2017*).

The cafeteria diet is a robust model of inducing dietary obesity in laboratory animals, promoting exacerbated hyperphagia and inflammation in a pronounced level (*Sampey et al., 2011*; *Oliva et al., 2019*). To date few studies have investigated the possible protective role of taurine within obesogenic diet models on pregnancy outcomes in rats. Therefore, the aim of this study was to examine whether taurine given with cafeteria diet prior to gestation, during gestation and lactation exerts any protective effects on dietary intakes, plasma circulating metabolites, amino acid profile and fetal growth and development until weaning.

## MATERIALS & METHODS

### Animals and diets

The experiments were performed under the license from the Ethics Committee of Hacettepe University, Ankara, Turkey, number: 2015/01. Animals were obtained from

Laboratory Animals Research and Application Centre. All animals were housed individually in plastic cages and subjected to a 12 h light-dark cycle at a temperature of 20–22 °C and 45% humidity. The animals were housed on wood shavings and had ad libitum access to food and water at all times. Animal care, feeding and maintaining of housing conditions were performed and checked by researchers and animal house personnel. After 1 week of habituation period, virgin female Wistar rats (aged 4 weeks) were randomly allocated to be fed either a control chow diet (CON; $n = 6$), control chow diet supplemented with 1.5% taurine in drinking water (*Li et al., 2013*; *Li et al., 2015*) (CONT; $n = 7$), cafeteria diet (CAF; $n = 7$) or cafeteria diet supplemented with 1.5% taurine in drinking water (CAFT; $n = 7$) for 8 weeks. In total, 27 female Wistar rats were used. Experimenters were not blind to treatment. Animals were then paired with a Wistar stud male and mating was confirmed by the appearance of a semen plug. The male rat was removed as soon as detecting semen plug. Animals continued to consume the same diets during pregnancy and lactation. Cafeteria diet consisted of control chow diet (%6 fat, %71 carbohydrate, %23 protein, energy 2800 kkal/kg, Kalecik, Kırıkkale, Turkey) with a random selection of highly energetic and palatable human foods. Data related to cafeteria diet were collected as previously described in *Buyukdere, Gulec & Akyol (2019)* and *Akyol, Langley-Evans & McMullen (2009)*. Briefly, the highly energetic and palatable human foods include biscuits, potato and corn crisps, milk chocolate, metro chocolate bar, kashar cheese, jelly candy, chocolate cake and peanuts. Four of these foods were given in a cup on the cage floor daily in excess quantities. In order to maintain variety two of these foods were replaced with new ones daily. Hence, rats did not have the same foods for more than two consecutive days at a time. All components of the cafeteria diet, including chow and water were individually weighed in and out of the cage between 09.00 and 10.00 h daily. Daily intakes of energy and macronutrients were calculated from the manufacturers' data, after allowing for weight changes due to drying of foods, as described previously (*Akyol, Langley-Evans & McMullen, 2009*). The animals were weighed daily during pre-mating, mating, birth and gestation periods. Weight loss of more than 15% of body weight, reduction in abilities of consuming diets and water or inability to walk properly were established as criteria for excluding animals prior to the planned end of the experiment but all of the animals successfully completed the study period. Thus, all analyses were performed through CON; $n = 6$, CONT; $n = 7$, CAF; $n = 7$ and CAFT; $n = 7$.

At birth, the birth weight, sex ratio and litter size were recorded. Litters were then culled to a maximum of eight pups (four males and four females, where possible). Body weights of all animals were recorded daily. Neonatal mortality was evaluated relative to litter size at birth and calculated by dividing number of dead pups to litter size (*Li et al., 2013*; *Li et al., 2015*). At the end of lactation, mothers and three male and three female offspring from each litter were culled using $CO_2$ asphyxia after overnight fasting. $CO_2$ asphyxia was applied by $CO_2$ euthanasia apparatus found in rat's home cages. Surviving animals were not allowed to see the procedure. Animals were exposed to 5.6 L/min $CO_2$ until complete cessation of breathing is observed for a minimum of 5 min. Cervical dislocation was performed to assure euthanasia. Body cavities of animals were opened and blood samples were taken by cardiac puncture and, major organs (liver, kidneys, heart, gonadal

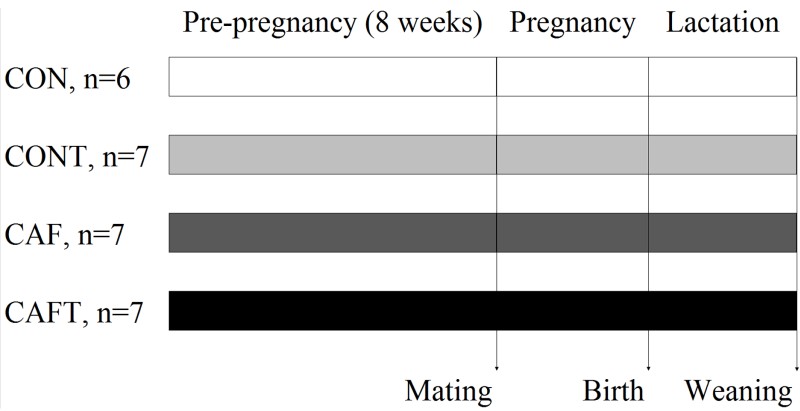

**Figure 1 Study design.** Control diet (CON); control diet with 1.5% taurine in drinking water (CONT); cafeteria diet (CAF); cafeteria diet with 1.5% taurine in drinking water (CAFT). Values for n show the number of successful pregnancies in each group.

and peri-renal adipose tissues) were dissected separately and weighed. The remaining offspring used in a different study. Figure 1 shows the study design. The study protocol was registered at the Scientific And Technological Research Council of Turkey, number 115S538.

## Plasma analyses

All blood samples were collected into heparinized capillary tubes and stored on ice until centrifuged in a hematocrit centrifuge. Plasma was collected and stored at −80 °C until required for analysis. Maternal plasma metabolite analyses were performed on fasting samples collected at weaning. Plasma leptin, adiponectin and insulin-like growth factor I (IGF-I) (R&D Systems, Inc., Minneapolis, MN, USA), insulin (Raybiotech, Norcross, GA, USA), total cholesterol and triglyceride (Hangzhou Eastbiopharm Co.,Ltd., Blue Ocean International Times Mansion, Hangzhou, China), c-peptide and HbA1C (Catalog No: E-EL-R0032, Elabscience, Houston, TX, USA) were measured by rat specific enzyme-linked immunosorbent assay. Plasma glucose and malondialdehyde were measured using commercially sourced assay kits (Cayman Chemical, Ann Arbor, MI, USA). Plasma taurine concentrations were analyzed using a Shimadzu LC-30 UHPLC system with auto sampler. Plasma amino acid analyses were performed using a GC amino acid kit (EZ:faast; Phenomenex, Torrance, CA, USA) (*Badawy, Morgan & Turner, 2008*). All assays were performed according to manufacturer's instructions and intra- and inter-assay coefficients of variation were <10%.

## Statistical analyses

All data were analyzed using the Statistical Package for Social Sciences (version 16; SPSS, Inc., Chicago, IL, USA). Data were tested for normality using Shapiro–Wilks normality test and visual analytical methods (histograms, probability plots). Data that did not meet the criteria required for parametric analysis were transformed to achieve normal distribution and equal variance. A general linear model analysis of variance (ANOVA) (fixed factors, maternal diet) was used for normally distributed data. Repeated-measures

**Table 1** Average daily maternal intakes of energy and nutrients per body weight during the pre-gestational, gestational and lactation periods.

| Period | Group | Dietary intake | | | | |
|---|---|---|---|---|---|---|
| | | Energy (kJ/g/d) | Fat (g/g/d)[†] | Protein (g/g/d)[§] | Carbohydrate (g/g/d)[‡] | Na (mg/g/d)[¥] |
| Pre-gestational intakes (weeks −7 to 0) | CON | 1.357 ± 0.060 | 0.002 ± 0.001[a] | 0.019 ± 0.001[a] | 0.058 ± 0.003[a,b] | 0.209 ± 0.011[a] |
| | CONT | 1.395 ± 0.056 | 0.002 ± 0.001[a] | 0.019 ± 0.001[a] | 0.059 ± 0.002[b] | 0.215 ± 0.011[a] |
| | CAF | 1.566 ± 0.056 | 0.016 ± 0.001[b] | 0.010 ± 0.001[b] | 0.049 ± 0.002[a,c] | 0.293 ± 0.011[b] |
| | CAFT | 1.503 ± 0.056 | 0.016 ± 0.001[b] | 0.010 ± 0.001[b] | 0.045 ± 0.002[c] | 0.285 ± 0.011[b] |
| Gestational intakes (weeks 1 to 3) | CON | 1.184 ± 0.057 | 0.002 ± 0.001[a] | 0.016 ± 0.000[a] | 0.050 ± 0.002[a] | 0.182 ± 0.013[a,b] |
| | CONT | 1.127 ± 0.053 | 0.002 ± 0.001[a] | 0.016 ± 0.000[a] | 0.048 ± 0.002[a] | 0.173 ± 0.012[a] |
| | CAF | 1.259 ± 0.053 | 0.013 ± 0.001[b] | 0.008 ± 0.000[b] | 0.037 ± 0.002[b] | 0.220 ± 0.012[b] |
| | CAFT | 1.185 ± 0.053 | 0.013 ± 0.001[b] | 0.007 ± 0.000[b] | 0.037 ± 0.002[b] | 0.227 ± 0.012[b] |
| Lactation intakes (weeks 4 to 6) | CON | 2.368 ± 0.141 | 0.004 ± 0.002[a] | 0.033 ± 0.002[a] | 0.101 ± 0.005[a] | 0.364 ± 0.039 |
| | CONT | 2.100 ± 0.130 | 0.003 ± 0.001[a] | 0.029 ± 0.001[a] | 0.089 ± 1.004[a] | 0.323 ± 0.036 |
| | CAF | 2.091 ± 0.130 | 0.023 ± 0.001[b] | 0.014 ± 0.001[b] | 0.54 ± 0.004[b] | 0.424 ± 0.036 |
| | CAFT | 2.379 ± 0.130 | 0.023 ± 0.001[b] | 0.015 ± 0.001[b] | 0.68 ± 0.004[b] | 0.459 ± 0.036 |

Notes:

Mean values with their standard errors, $n = 6$ (CON), $n = 7$ (CONT, CAF and CAFT). CON, control chow diet; CONT, control chow diet supplemented with taurine; CAF, cafeteria diet; CAFT, cafeteria diet supplemented with taurine.

[†] Diet and study weeks significantly influenced fat intake during pre-gestation (Diet, $P < 0.001$; study weeks, $P < 0.001$), gestation (Diet, $P < 0.001$; study weeks, $P = 0.001$) and lactation (Diet, $P < 0.001$; study weeks, $P < 0.001$). A significant interaction between diet and study weeks also influenced fat intake during pre-gestation, gestation and lactation ($P < 0.05$).

[§] Diet and study weeks significantly influenced protein intake during pre-gestation (Diet, $P < 0.001$; study weeks, $P < 0.001$), gestation (Diet, $P < 0.001$; study weeks, $P < 0.001$) and lactation (Diet, $P < 0.001$; study weeks, $P < 0.001$). A significant interaction between diet and study weeks also influenced protein intake during gestation and lactation ($P < 0.05$).

[‡] Diet and study weeks significantly influenced carbohydrate intake during pre-gestation (Diet, $P = 0.001$; study weeks, $P < 0.001$), gestation (Diet, $P < 0.001$; study weeks, $P < 0.001$) and lactation (Diet, $P < 0.001$; study weeks, $P < 0.001$). A significant interaction between diet and study weeks also influenced carbohydrate intake during gestation and lactation ($P < 0.01$).

[¥] Diet and study weeks significantly influenced sodium intake during pre-gestation (Diet, $P < 0.001$; study weeks, $P < 0.001$) and gestation (Diet, $P = 0.008$; study weeks, $P = 0.021$). A significant interaction between diet and study weeks also influenced sodium intake during pre-gestation and lactation ($P < 0.05$).

[a,b,c] Mean values with unlike superscript letters were significantly different ($P < 0.05$).

ANOVA was performed to compare the mean differences between groups in those parameters measured at different time points (for example, weekly body weights and energy intakes). The effect of maternal diet on fetal outcomes was assessed using a general linear model analysis of variance (ANOVA) (fixed factors, maternal diet and sex). Post hoc testing (Tukey's test) was applied for the significant main effects of the diet. Data that did not normally distribute was analysed using the non-parametric Kruskal–Wallis test and comparisons between two groups were performed with the Mann-Whitney U-test. The proportion of neonatal deaths per litter was analyzed via Chi-square test. Values are expressed as mean values with their standard errors. $p < 0.05$ was considered statistically significant. Power analysis indicated that 6 animals per group was sufficient to detect a minimum 12% energy intake difference with a power of 80% and alpha 0.05 (*Akyol, Langley-Evans & McMullen, 2009*). The study data is submitted as a Supplementary File.

## RESULTS

During pre-gestational period energy intakes of dams did not differ between groups ($P = 0.062$) (Table 1). CAF and CAFT fed animals had significantly higher intakes of fat ($P < 0.001$), and Na ($P < 0.001$), and significantly lower intakes of protein ($P < 0.001$). While carbohydrate intake of CAF and CAFT fed animals was significantly lower than

CONT fed animals ($P < 0.05$), CAF fed animals exhibited a similar carbohydrate intake in comparison to CON fed animals ($P = 0.105$).

There were no statistically significant differences in energy intake between the groups during gestation and lactation ($P > 0.05$). CAF and CAFT had significantly higher fat ($P < 0.001$) and Na ($P < 0.001$), and significantly lower protein ($P < 0.001$) and carbohydrate ($P < 0.001$) intake during gestation (Table 1). Alike pre-gestational and gestational periods, fat ($P < 0.001$) intake was significantly higher in CAF and CAFT whereas protein ($P < 0.001$) and carbohydrate ($P < 0.001$) intakes were significantly lower during the lactation period. A significant interaction between diet and study weeks also influenced fat and protein intake during gestation and lactation ($P < 0.05$).

Since taurine was added to drinking water in CONT and CAFT, water intake of animals was also recorded (Fig. 2A). During the pre-gestational period CAF (28.01 ± 0.82 g/day) and CAFT (27.81 ± 0.82 g/day) consumed significantly lower amount of water than CON (41.31 ± 0.88 g/day) and CONT (45.77 ± 0.82 g/day) ($P < 0.001$). During the gestation and lactation periods a similar pattern was observed in groups as CAF (Gestation: 29.72 ± 1.44 g/day, lactation: 42.16 ± 2.94 g/day) and CAFT (Gestation: 29.85 ± 1.44 g/day, lactation: 42.07 ± 2.94 g/day) consumed significantly lower amount of water in comparison to CON (Gestation: 51.54 ± 1.56 g/day, lactation: 94.84 ± 3.18 g/day) and CONT (Gestation: 55.15 ± 1.44 g/day, lactation: 101.89 ± 2.94 g/day) ($P < 0.001$). Estimated taurine intake of CAFT dams (2.52 ± 0.19 mg/g body weight/day) was significantly lower than CONT dams (3.95 ± 0.19 mg/g body weight/day) during the pre-gestational period ($P < 0.001$) (Fig. 2B). During the gestation and lactation periods a similar pattern was observed in CAFT group (Gestation: 1.68 ± 0.14 mg/g body weight/day, lactation: 2.86 ± 0.28 mg/g body weight/day) as estimated taurine consumption of CAFT dams was significantly lower than CONT dams (Gestation: 3.11 ± 0.14 mg/g body weight/day, lactation: 5.47 ± 0.28 mg/g body weight/day) ($P < 0.001$).

The body weights of the dams did not vary significantly at the beginning of the experiment. All animals gained weight during the pre-pregnancy and pregnancy periods (Fig. 3). A significant interaction between diet and study weeks influenced average body weights during pre-gestational period (CON: 168.05 ± 1.69 g, CONT: 179.99 ± 1.69 g, CAF: 178.77 ± 1.57 g and CAFT: 175.62 ± 1.57 g; interaction between diet*study weeks $P = 0.002$). During the pregnancy period, all groups exhibited similar gestational body weights (CAF: 272.08 ± 4.78 g, CAFT: 268.21 ± 4.78 g, CON: 252.22 ± 5.16 g, CONT: 267.88 ± 4.78 g, $P = 0.369$). Weight gain slowed in rats fed the CAFT (243.48 ± 4.58 g) diet during lactation in comparison with those fed the CONT (276.14 ± 4.58 g) ($P = 0.035$). Both CAF and CAFT groups exhibited weight loss during lactation.

Dietary treatment during pre-gestational, gestational and lactation periods significantly influenced maternal liver, kidney and adipose tissue weights at the end of the lactation period (Table 2). Taurine addition to cafeteria diet or chow diet did not alter liver weight between CON and CONT or CAF and CAFT. However, CAF and CAFT had significantly lower liver weights than CON and CONT ($P < 0.001$). Both of the right and left kidney weights exhibited a similar pattern as CAF and CAFT had significantly lower kidney

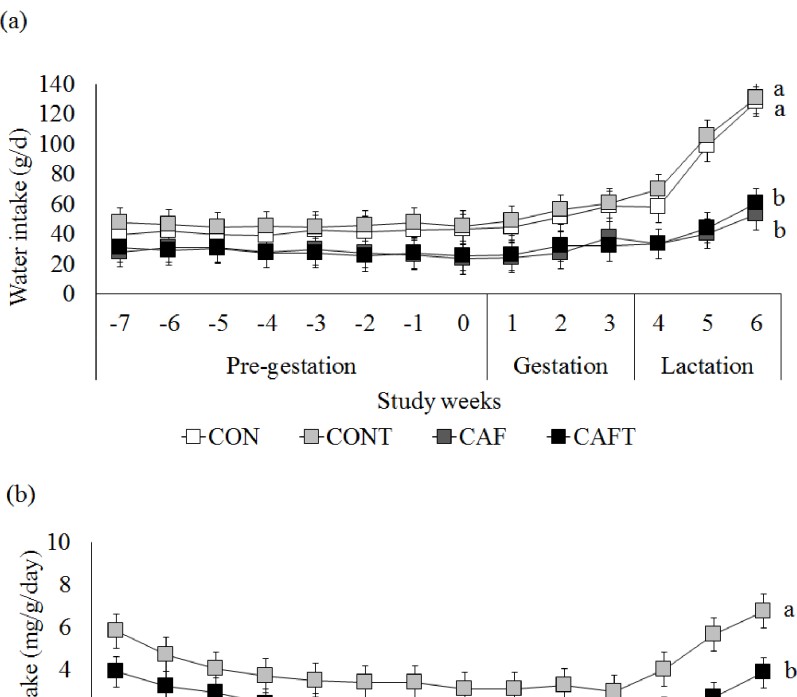

(a)

(b)

**Figure 2 Maternal water and taurine intakes.** (A) Water intake before and during pregnancy in rats fed a CON, control chow diet; CONT, control chow diet supplemented with taurine; CAF, cafeteria diet; CAFT, cafeteria diet supplemented with taurine. Values are means, with standard errors represented by vertical bars, for $n = 6$ (CON); $n = 7$ (CONT, CAF and CAFT). Water intake during the pre-gestational period was significantly lower in the CAF and CAFT fed animals (effect of diet, $P < 0.001$). Water intake during the gestational period was significantly lower in the CAF and CAFT fed animals (effect of diet and study weeks, $P < 0.001$). Water intake during the lactation period was significantly lower in the CAF and CAFT fed animals (effect of diet, $P < 0.001$; study weeks $P < 0.001$; interaction between diet*study weeks, $P < 0.001$). (B) Taurine intake (mg/g body weight/day) before and during pregnancy in rats fed a CONT, control chow diet supplemented with taurine; CAFT, cafeteria diet supplemented with taurine. Values are means, with standard errors represented by vertical bars, for $n = 7$ (CONT and CAFT). Taurine intake during the pre-gestational period was significantly lower in the CAFT fed animals (effect of diet, $P < 0.001$). Taurine intake during the gestational period was significantly lower in the CAFT fed animals (effect of diet and study weeks, $P < 0.001$). Taurine intake during the lactation period was significantly lower in the CAFT fed animals (effect of diet, $P < 0.001$; study weeks $P < 0.001$; interaction between diet*study weeks, $P < 0.001$). [a,b]Mean values with unlike superscript letters were significantly different ($P < 0.001$).

weights than CON and CONT ($P < 0.05$). Fat depot mass data indicated that taurine addition to cafeteria diet did not exert a protective effect on gonadal and peri-renal fat mass since CAF and CAFT had significantly heavier values than CON and CONT ($P < 0.001$) and there were no significant difference between CAF and CAFT groups.

Maternal plasma glucose, insulin, IGF-1, C-peptide, cholesterol and triglyceride were unaffected by dietary treatment at the end of the lactation (Table 3). CONT, CAF and

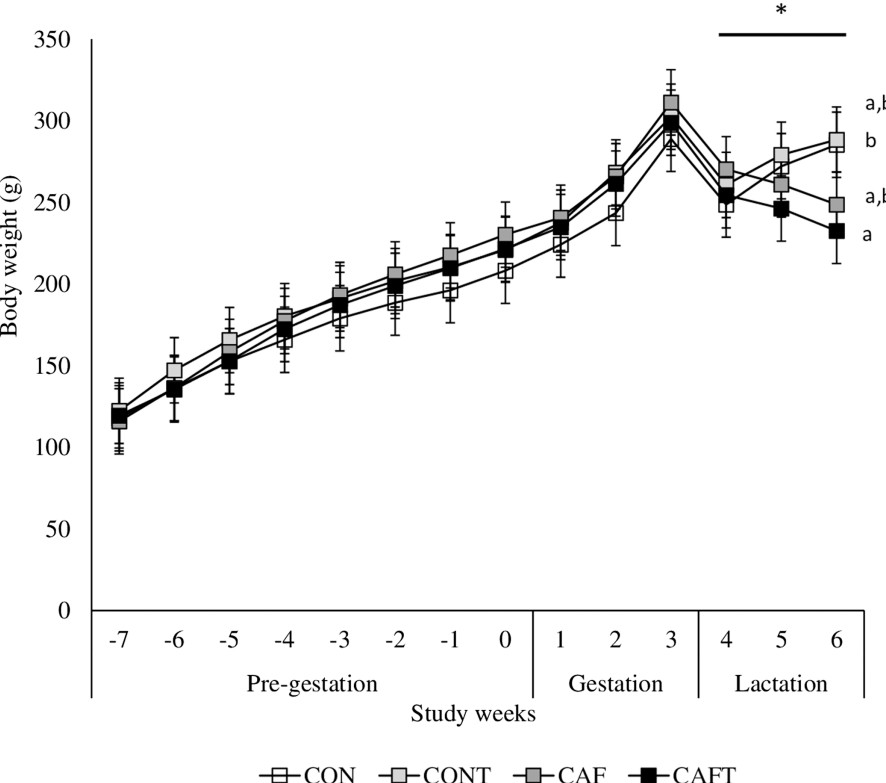

**Figure 3 Body weight changes before and during pregnancy.** CON, control chow diet; CONT, control chow diet supplemented with taurine; CAF, cafeteria diet; CAFT, cafeteria diet supplemented with taurine. Values are means, with standard errors represented by vertical bars, for $n = 6$ (CON); $n = 7$ (CONT, CAF and CAFT). A significant interaction between diet and study weeks influenced body weight during pre-gestational period (interaction between diet*study weeks, $P = 0.002$). Body weight during the gestational period did not differ between groups (effect of diet, $P = 0.369$; study weeks, $P < 0.001$). Body weight during the lactation period was significantly lower in the CAFT fed animals than CONT fed animals (effect of diet, $P = 0.035$; interaction between diet*study weeks, $P < 0.001$). [a,b] Mean values with unlike superscript letters were significantly different ($P < 0.001$).

**Table 2 Maternal organ weight and fat depot mass.**

| Group | Organ weight or fat depot mass | | | | | |
|---|---|---|---|---|---|---|
| | Liver[*] | Heart | Right kidney[§] | Left kidney[‡] | Gonadal fat[¥] | Peri-renal fat[¶] |
| CON | $10.99 \pm 0.59^a$ | $0.72 \pm 0.03$ | $1.13 \pm 0.05^a$ | $1.05 \pm 0.04^a$ | $1.67 \pm 0.52^a$ | $0.58 \pm 0.22^a$ |
| CONT | $11.39 \pm 0.55^a$ | $0.76 \pm 0.03$ | $1.15 \pm 0.04^a$ | $1.11 \pm 0.04^a$ | $1.67 \pm 0.48^a$ | $0.58 \pm 0.20^a$ |
| CAF | $8.58 \pm 0.55^b$ | $0.77 \pm 0.03$ | $0.82 \pm 0.04^b$ | $0.76 \pm 0.04^b$ | $5.92 \pm 0.48^b$ | $2.53 \pm 0.22^b$ |
| CAFT | $8.12 \pm 0.55^b$ | $0.74 \pm 0.03$ | $0.87 \pm 0.04^b$ | $0.82 \pm 0.04^b$ | $4.42 \pm 0.48^b$ | $2.10 \pm 0.24^b$ |

Notes:
Mean values with their standard errors, $n = 6$ (CON), $n = 7$ (CONT, CAF and CAFT). CON, control chow diet; CONT, control chow diet supplemented with taurine; CAF, cafeteria diet; CAFT, cafeteria diet supplemented with taurine.
[*] Diet significantly influenced liver weight ($P < 0.001$, ANOVA).
[§] Diet significantly influenced right kidney weight ($P < 0.001$, Kruskal–Wallis).
[‡] Diet significantly influenced left kidney weight ($P < 0.001$, Kruskal–Wallis).
[¥] Diet significantly influenced gonadal fat mass ($P < 0.001$, ANOVA).
[¶] Diet significantly influenced peri-renal fat mass ($P < 0.001$, ANOVA).
[a,b] Mean values with unlike superscript letters were significantly different ($P < 0.05$).

**Table 3 Concentrations of biochemical parameters in maternal plasma.**

| Biochemical Parameter | Study Groups | | | |
|---|---|---|---|---|
| | CON | CONT | CAF | CAFT |
| Glucose (mg/dL) | 130.56 ± 11.23 | 159.41 ± 10.40 | 123.78 ± 11.23 | 121.74 ± 10.40 |
| Insulin (μU/mL) | 41.34 ± 4.77 | 30.51 ± 4.41 | 26.06 ± 4.77 | 27.79 ± 4.41 |
| IGF-1 (ng/mL) | 745.21 ± 170.80 | 963.88 ± 158.13 | 1,358.57 ± 170.80 | 773.23 ± 158.13 |
| C-peptide (ng/mL) | 2.22 ± 0.23 | 5.36 ± 1.21 | 4.67 ± 1.23 | 7.07 ± 1.21 |
| HbA1c[¶] (ng/mL) | 12.06 ± 1.80[a] | 20.84 ± 1.81[b] | 27.19 ± 1.80[b] | 20.30 ± 1.85[b] |
| Cholesterol (mmol/L) | 2.89 ± 0.15 | 2.57 ± 0.14 | 2.76 ± 0.15 | 2.59 ± 0.14 |
| Triglycerides (mmol/L) | 1.49 ± 0.18 | 1.56 ± 0.19 | 1.55 ± 0.19 | 1.31 ± 1.18 |
| Leptin (ng/mL)[*] | 0.55 ± 0.09[a] | 0.63 ± 0.09[a] | 0.93 ± 0.09[b] | 0.76 ± 0.09[a,b] |
| Adiponectin (μg/mL)[†] | 3.09 ± 1.58[a] | 3.52 ± 1.46[a] | 11.53 ± 1.58[b] | 15.73 ± 1.58[b] |
| Malondialdehyde (μM)[§] | 11.76 ± 4.48[a] | 21.74 ± 4.15[a] | 32.69 ± 4.48[b] | 16.86 ± 4.15[a] |

Notes:
Mean values with their standard errors, $n = 6$ (CON), $n = 7$ (CONT, CAF and CAFT). CON, control chow diet; CONT, control chow diet supplemented with taurine; CAF, cafeteria diet; CAFT, cafeteria diet supplemented with taurine.
[¶] Diet significantly influenced maternal plasma HbA1c levels ($P = 0.02$, Kruskal–Wallis)
[*] Diet significantly influenced maternal plasma leptin levels ($P = 0.047$, ANOVA).
[†] Diet significantly influenced maternal plasma adiponectin levels ($P < 0.001$, Kruskal–Wallis).
[§] Diet significantly influenced maternal plasma malondialdehyde levels ($P = 0.02$, ANOVA).
[a,b] Mean values with unlike superscript letters were significantly different ($P < 0.05$, ANOVA).

CAFT displayed higher levels of HbA1c than CON. Leptin was significantly higher in CAF compared to CON and CONT whereas it was similar to all other three groups in CAFT. Adiponectin was significantly higher in CAF and CAFT than CON and CONT. Malondialdehyde was significantly higher only in CAF in comparison to CON and CONT. Maternal plasma taurine was increased in CONT compared to CON, CAF and CAFT (Table 4). Also, taurine was significantly lower in CAF than CON, CONT and CAFT. CAF and CAFT exhibited higher levels of serine and lower levels of tyrosine than CON. Phenylalanine was lower and aspartic acid was higher in CAFT than CON.

Litter size (CON: 11.00 ± 1.19, CONT: 10.43 ± 1.11, CAF: 9.57 ± 1.11, CAFT: 9.57 ± 1.11, $P = 0.775$) and sex ratio (male:female) (CON: 0.76 ± 0.49, CONT: 1.22 ± 0.46, CAF: 1.56 ± 0.46, CAFT: 1.79 ± 0.46, $P = 0.464$) were similar between the study groups. Similarly, birth weights of pups did not differ between groups ($p = 0.532$) but male offspring's birth weights were higher than female offspring (CON male: 5.87 ± 0.13 g, female: 5.40 ± 0.10 g; CONT male: 5.70 ± 0.11 g, female: 5.36 ± 0.11 g; CAF male: 5.64 ± 0.11 g, female: 5.30 ± 0.12 g and CAFT male: 5.64 ± 0.12 g, female: 5.37 ± 0.11 g, $P < 0.001$). Maternal taurine supplementation significantly increased the proportion of neonatal deaths per litter only in the CONT group and sex of the animals had no effect on this outcome (CONT: 12.3% versus CON: 3.0%, CAF: 4.5% and CAFT: 1.5%, $P = 0.023$). Those offspring died during the first week of lactation.

Offspring weight gain was significantly influenced by maternal diet ($P < 0.001$) and study weeks ($P < 0.001$) during lactation but there was no effect of sex (Fig. 4). Maternal diet and study weeks exhibited a significant interaction ($P < 0.001$). Although birth weight did not vary between the groups, this situation disappeared over time and CAF and CAFT offspring displayed lower body weights in comparison to CON and CONT

**Table 4 Concentrations of amino acids in maternal plasma.**

**Amino acids (μmol/L)**

| | CON | CONT | CAF | CAFT |
|---|---|---|---|---|
| Alanine | 1,395.82 ± 123.51 | 1,286.27 ± 114.35 | 1,389.92 ± 123.51 | 1,196.91 ± 114.35 |
| Asparagine | 104.54 ± 8.01 | 81.49 ± 7.42 | 85.43 ± 8.01 | 82.59 ± 7.42 |
| Aspartic acid[*] | 884.70 ± 93.41[a] | 931.75 ± 86.48[a] | 1,226.15 ± 93.41[a,b] | 1,473.46 ± 86.48[b] |
| Phenylalanine[†] | 1,047.72 ± 70.25[a] | 943.44 ± 65.04[a,b] | 831.10 ± 70.25[a,b] | 775.27 ± 65.04[b] |
| Glycine | 721.94 ± 58.33 | 774.16 ± 53.99 | 723.66 ± 58.33 | 589.13 ± 53.99 |
| Glutamic acid | 370.66 ± 65.59 | 362.06 ± 60.73 | 436.48 ± 65.59 | 483.81 ± 60.73 |
| Glutamine | 328.56 ± 84.10 | 276.20 ± 77.86 | 383.76 ± 84.10 | 306.11 ± 77.86 |
| Histidine | 148.90 ± 15.34 | 149.09 ± 14.21 | 138.04 ± 15.34 | 165.70 ± 14.21 |
| Isoleucine | 177.64 ± 14.63 | 152.56 ± 13.54 | 147.57 ± 14.63 | 146.25 ± 13.54 |
| Lysine | 1,845.06 ± 166.31 | 1,911.47 ± 153.97 | 1,826.96 ± 166.31 | 2,216.45 ± 153.97 |
| Leucine | 358.83 ± 27.33 | 308.95 ± 25.30 | 269.25 ± 27.33 | 272.11 ± 25.30 |
| Methionine | 201.70 ± 28.16 | 205.31 ± 26.07 | 164.34 ± 28.16 | 169.19 ± 26.07 |
| Ornithine | 306.06 ± 49.88 | 313.74 ± 46.18 | 261.22 ± 49.88 | 181.25 ± 46.18 |
| Proline | 283.99 ± 22.29 | 264.72 ± 20.64 | 319.95 ± 22.29 | 299.37 ± 20.64 |
| Serine[§] | 623.55 ± 124.60[a] | 548.92 ± 115.36[a] | 1,288.82 ± 124.60[b] | 1,277.52 ± 115.36[b] |
| Cystine | 24.24 ± 4.34 | 26.32 ± 5.02 | 16.80 ± 3.89 | 16.63 ± 3.55 |
| Tyrosine[‡] | 147.64 ± 9.42[a] | 127.09 ± 8.72[a] | 103.73 ± 9.42[b] | 98.54 ± 8.72[b] |
| Threonine | 496.53 ± 44.89 | 405.01 ± 41.56 | 414.88 ± 44.89 | 435.25 ± 41.56 |
| Tryptophan | 139.10 ± 16.31 | 169.02 ± 15.10 | 107.18 ± 16.31 | 121.74 ± 15.10 |
| Valine | 354.81 ± 29.85 | 327.37 ± 27.64 | 275.34 ± 29.85 | 299.11 ± 27.64 |
| Cystathionine | 22.96 ± 2.96 | 26.11 ± 2.96 | 18.69 ± 2.96 | 21.07 ± 2.74 |
| Alpha-aminoadipic acid | 967.51 ± 374.54 | 327.64 ± 346.75 | 2,338.30 ± 374.54 | 555.52 ± 346.75 |
| Taurine[¶] | 287.16 ± 44.49[a] | 473.24 ± 41.19[c] | 151.83 ± 44.49[b] | 291.16 ± 41.19[a] |

Notes:
Mean values with their standard errors, $n = 6$ (CON), $n = 7$ (CONT, CAF and CAFT). CON, control chow diet; CONT, control chow diet supplemented with taurine; CAF, cafeteria diet; CAFT, cafeteria diet supplemented with taurine.
[*] Diet significantly influenced plasma aspartic acid concentrations ($P < 0.001$, ANOVA).
[†] Diet significantly influenced plasma phenylalanine concentrations ($P = 0.045$, ANOVA).
[§] Diet significantly influenced plasma serine concentrations ($P < 0.001$, ANOVA).
[‡] Diet significantly influenced plasma tyrosine concentrations ($P = 0.008$, Kruskal–Wallis).
[¶] Diet significantly influenced plasma taurine concentrations ($P < 0.001$, Kruskal–Wallis).
[a,b,c] Mean values with unlike superscript letters were significantly different ($P < 0.05$).

offspring in both genders (CON, male: 19.00 ± 0.60 g, female: 18.88 ± 0.47 g; CONT, male: 20.31 ± 0.55 g, female: 20.44 ± 0.50 g; CAF, male: 16.91 ± 0.50 g, female: 16.95 ± 0.53 g; CAFT male: 16.34 ± 0.48 g, female: 16.56 ± 0.49 g, $P < 0.001$). In addition, maternal diet significantly influenced liver, brain, kidney and heart weights of offspring in both genders at the end of lactation (Table 5). Fetal exposure to cafeteria diet led to lower liver, brain and kidney weights compared with CON and CONT groups ($P < 0.001$). Liver, brain and kidney weights of CAFT offspring exhibited similar patterns like CAF offspring ($P < 0.001$). Furthermore, a reduction in heart weight was observed in the offspring of CAFT ($P = 0.011$).

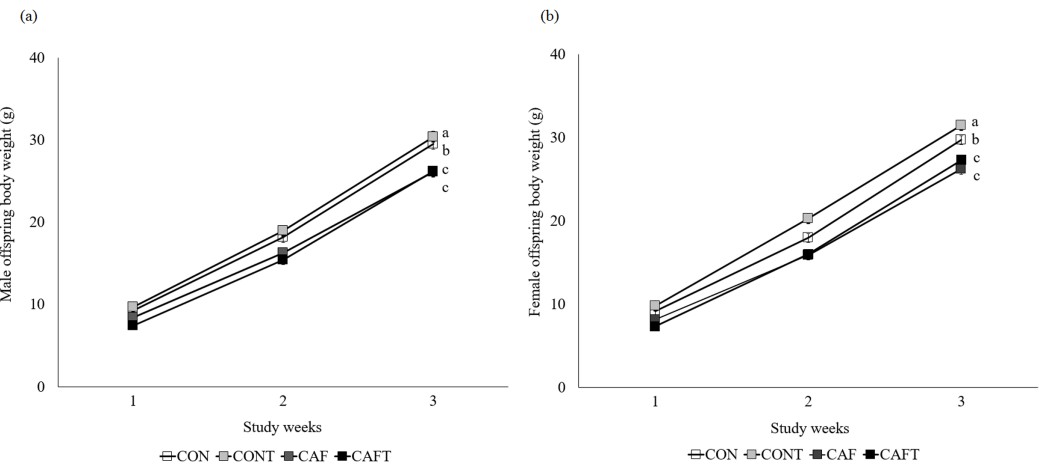

**Figure 4 Body weight changes of offspring during lactation period.** CON, rats fed the control chow diet before and during pregnancy; CONT, rats fed the control chow diet supplemented with taurine before and during pregnancy; CAF, rats fed the cafeteria diet before and during pregnancy; CAFT, rats fed the cafeteria diet supplemented with taurine before and during pregnancy. (A) Body weight changes of male offspring during lactation period. (B) Body weight changes of female offspring during lactation period. Values are means, with standard errors represented by vertical bars, for $n = 18$ (CON, males), $n = 28$ (CON, female), $n = 20$ (CONT, male), $n = 25$ (CONT, female), $n = 25$ (CAF, male), $n = 22$ (CAF, female), $n = 27$ (CAFT, male) and $n = 26$ (CAFT, female). Body weight was significantly influenced by maternal diet ($P < 0.001$), study weeks ($P < 0.001$) and interaction of maternal diet and study weeks ($P < 0.001$). Body weight during the lactation period was significantly lower in CAF and CAFT offspring than CON and CONT offspring (effect of diet, $P < 0.001$). [a,b,c] Mean values with unlike superscript letters were significantly different ($P < 0.001$)

**Table 5 Organ weight of offspring at the end of lactation.**

| Sex | Group | Liver[*] | Brain[†] | Right kidney[‡] | Left kidney[¥] | Heart[§] |
|---|---|---|---|---|---|---|
| Male | CON | 1,637 ± 0,074[a] | 1,344 ± 0,020[a] | 0,250 ± 0,010[a] | 0,237 ± 0,009[a] | 0,192 ± 0,008[a] |
| | CONT | 1,488 ± 0,069[a] | 1,351 ± 0,018[a] | 0,225 ± 0,009[a] | 0,211 ± 0,009[a] | 0,195 ± 0,008[a] |
| | CAF | 1,144 ± 0,061[b] | 1,265 ± 0,016[b] | 0,182 ± 0,008[b] | 0,169 ± 0,008[b] | 0,190 ± 0,007[a] |
| | CAFT | 1,111 ± 0,058[b] | 1,275 ± 0,015[b] | 0,177 ± 0,008[b] | 0,165 ± 0,007[b] | 0,163 ± 0,007[b] |
| Female | CON | 1,574 ± 0,055[a] | 1,307 ± 0,014[a] | 0,241 ± 0,007[a] | 0,226 ± 0,007[a] | 0,187 ± 0,006[a] |
| | CONT | 1,482 ± 0,062[a] | 1,304 ± 0,016[a] | 0,234 ± 0,008[a] | 0,227 ± 0,008[a] | 0,199 ± 0,007[a] |
| | CAF | 1,169 ± 0,067[b] | 1,198 ± 0,017[b] | 0,186 ± 0,009[b] | 0,182 ± 0,007[b] | 0,187 ± 0,008[a] |
| | CAFT | 1,172 ± 0,059[b] | 1,262 ± 0,016[b] | 0,192 ± 0,008[b] | 0,184 ± 0,008[b] | 0,175 ± 0,007[b] |

Notes:

Mean values with their standard errors, $n = 12$ (CON, males), $n = 22$ (CON, female), $n = 14$ (CONT, male), $n = 17$ (CONT, female), $n = 18$ (CAF, male), $n = 15$ (CAF, female), $n = 20$ (CAFT, male) and $n = 19$ (CAFT, female). CON, control chow diet; CONT, control chow diet supplemented with taurine; CAF, cafeteria diet; CAFT, cafeteria diet supplemented with taurine.

[*] Maternal diet significantly influenced liver weight ($P < 0.001$, ANOVA).
[†] Maternal diet and sex of the animals significantly influenced brain weight (Maternal Diet, $P = 0.001$, ANOVA).
[‡] Maternal diet significantly influenced right kidney weight ($P < 0.001$, ANOVA).
[¥] Maternal diet significantly influenced left kidney weight (Maternal Diet, $P < 0.001$, ANOVA).
[§] Maternal diet significantly influenced heart weight ($P < 0.001$, ANOVA).
[a,b] Mean values with unlike superscript letters were significantly different ($P < 0.05$, ANOVA).

## DISCUSSION

The influence of maternal dietary strategies to prevent the development of chronic diseases on the developing offspring is relatively unknown in contrast to the more direct

inferences of neonatal health. Few animal studies support the notion that taurine supplementation may trigger differences in metabolic functions and physiology (*Rashid, Das & Sil, 2013*; *Zheng et al., 2017*). However, it is not clear whether such effects can continue on throughout pre-pregnancy, pregnancy and lactation. The maternal metabolic health during these periods is an important determinant of health status of offspring. Both animal and human studies revealed that gestation period does not terminate with birth but with end of lactation (*Stuebe & Rich-Edwards, 2009*). Hence, the primary aim of this study was to compare the effect of a maternal cafeteria diet with taurine supplemented cafeteria diet in terms of pre-pregnancy nutritional status, pregnancy progression, outcomes and fetal growth and development throughout lactation. In this regard, firstly, the present study showed significant nutritional and metabolic changes in dams in response to a maternal cafeteria diet at weaning. More specifically, maternal taurine supplementation reversed some of these changes at a modest level through preventing maternal hyperleptinemia, reducing malondialdehyde and increased plasma taurine levels. Furthermore, increased neonatal mortality was observed in taurine supplemented-control group.

Previous studies reported that ingestion of the cafeteria diet before pregnancy led to hyperphagia and increased energy intake (*Akyol, Langley-Evans & McMullen, 2009*; *Crew, Waddell & Mark, 2016*; *Sánchez-Blanco et al., 2016*). Unlike this finding, energy intake of cafeteria fed dams did not differ from other groups prior to gestation in this study. One explanation might be the expression of the data was normalized to body weight in the current study. Furthermore, it was suggested that providing more food items (for example 40 highly palatable energy-dense human food) made the cafeteria diet more successful at sustained hyperphagia and greater weight gain (*George et al., 2019*). *Gugusheff et al. (2016)* reported similar energy intakes between cafeteria and control dams throughout pre-gestation and gestation periods which may be associated with presenting less food items in cafeteria diet. Also, energy intake of cafeteria fed dams did not differ from other groups during gestation in the current study. Similarly, one study reported that energy intakes of cafeteria fed dams were similar to other groups (*Ferro Cavalcante et al., 2014*) whereas other studies showed higher energy intakes during gestation in cafeteria group (*Akyol, Langley-Evans & McMullen, 2009*; *Sánchez-Blanco et al., 2016*; *Vithayathil et al., 2018*). The present study demonstrated that energy intake was not different between groups during lactation, which has been reported previously (*Ferro Cavalcante et al., 2014*), but not in all studies (*Bayol, Farrington & Stickland, 2007*; *Speight et al., 2017*; *Vithayathil et al., 2018*). These contradictory results may be due to type of foods used in the cafeteria diet, differences in the duration and timing of intervention. Cafeteria diet is an unbalanced diet with a higher percentage of total energy coming from fat and a lower percentage coming from carbohydrates and proteins compared to control diet (*Sampey et al., 2011*). Thus, both CAF and CAFT groups consumed lower protein and carbohydrate and remarkably greater fat compared to CON and CONT groups. Overall, taurine supplementation did not affect food consumption, energy intake and food preferences during gestation and lactation. This is consistent with the findings of previous studies (*Li et al., 2013*; *Li et al., 2015*).

Since the total energy intake of dams fed the cafeteria diet was similar to dams fed the control diet throughout pre-gestation, gestation and lactation, total body weights did not differ between cafeteria and control groups at the end of lactation. Some studies reported that consumption of cafeteria diet during 8 weeks before gestation might fail to trigger significant weight gain (*Jacobs et al., 2014*; *Rossetti et al., 2020*). *Rossetti et al. (2020)* indicated that they maintained feeding cafeteria diet until 14th week of the experiment before mating in order to detect significantly increased body weights in CAF group. Therefore, differences in the duration may lead to contradictory results regarding pre-gestational body weights. Also, at mating no significant differences were observed regarding body weight between groups but previous data showed that at day 20 of gestation cafeteria diet fat dams had greater body fat accumulation which is an essential component of obesity (*Akyol, Langley-Evans & McMullen, 2009*). In a previous study, profound adiposity was showed although no difference was observed in terms of body weight between cafeteria and control groups (*Buyukdere, Gulec & Akyol, 2019*). Furthermore, the organ weight data at the end of lactation showed that CAF and CAFT animals had significantly increased gonadal and peri-renal fat depots (72% increase in gonadal fat and 77% in peri-renal fat in CAF). Therefore, it can be suggested that cafeteria diet led to increased adiposity at the end of 8 weeks of pre-gestational feeding and this model produced an efficient model of maternal obesity.

Ingestion of cafeteria diet resulted in weight loss during lactation and this was not influenced by taurine supplementation. This can be explained by suggesting that dams fed the cafeteria diet during lactation could invest more energy to milk production and hence their milk could be richer than the milk produced by the chow diet-fed lactating rats (*Bayol, Farrington & Stickland, 2007*). Indeed, it was demonstrated that milk from cafeteria diet-fed dams contained higher concentration of fat and lower concentration of protein when compared to controls. This may explain the reduced growth rate of the offspring of cafeteria diet fed dams (*Pomar et al., 2017*). It can be suggested that maternal ingestion of cafeteria diet affected the offspring in a similar setting to the effects of a low protein diet. Lower protein content of the maternal diet could lead to lower body weight at weaning (*Bayol, Farrington & Stickland, 2007*; *Pomar et al., 2017*). In the present study, both male and female CAF and CAFT offspring were leaner than CON and CONT offspring at weaning. In studies conducted with low protein diet models supplementation with taurine did not prevent growth retardation of the offspring during lactation (*Boujendar et al., 2002*; *Merezak et al., 2004*). However, another study reported that weaning weights of offspring exposed to obesogenic diet supplemented with taurine was similar to control offspring (*Li et al., 2015*). Different dietary exposure models could lead to differential effects on fetal growth and development as protein content of obesogenic diet models exhibited distinct levels of protein.

Despite weight loss during lactation, cafeteria diet fed dams exhibited markedly increased adiposity at the end of lactation. Although CAFT group had lower gonadal and peri-renal fat depots than CAF group; this did not reach to statistical significance in the current study. Thus, taurine supplementation did not decrease gonadal and peri-renal fat pad weights. Supplementation of 5% (wt/wt) taurine (estimated taurine intake 3 mg/g

body weight/day and 11 mg/g body weight/day respectively) was shown to prevent tissue fat accumulation and obesity with increased energy expenditure (*Tsuboyama-Kasaoka et al., 2006*; *Lin et al., 2013*), while high fat diet induced obesity in mice could not be prevented by 1% taurine treatment (estimated taurine intake 1.7 mg/g body weight/day) (*Murakami, Kondo & Nagate, 2000*). This difference between the studies can be attributable to supplementation dosage and amount of estimated taurine intake. In the current study, estimated taurine intake of CAFT dams was 2.4 mg/g body weight/day. This result may suggest that higher amount of taurine is required to observe anti-obesogenic effects.

Plasma taurine levels were shown to decrease in obesity since taurine synthesis in white adipose tissue is reduced (*Tsuboyama-Kasaoka et al., 2006*). Also, *Rosa et al. (2014)* reported that obese women had lower plasma taurine levels than normal weight control group. However, *Li et al. (2013*, *2015)* indicated that dams fed with maternal obesogenic diets displayed similar plasma taurine concentrations to control dams. These differences might be due to the degree of fat deposition. CAF fed dams exerted decreased plasma taurine levels in this study. Most studies demonstrated that taurine supplementation improved plasma taurine levels in the setting of diet induced obesity (*Tsuboyama-Kasaoka et al., 2006*; *Li et al., 2013*; *Li et al., 2015*). In addition, taurine supplementation resulted in a marked elevation of taurine concentrations in CONT and CAFT dams in the current study.

Many studies have attempted to investigate the influence of maternal cafeteria diet on plasma glucose, insulin, triglyceride and total cholesterol levels. Some have found no differences in glucose and insulin (*Crew, Waddell & Mark, 2016*; *Crew et al., 2018*), triglyceride (*Mucellini et al., 2014*) and total cholesterol (*Jacobs et al., 2014*) concentrations as observed in the present study, while others have reported higher glucose and insulin (*Holemans et al., 2004*; *Bouanane et al., 2009*) cholesterol (*Mucellini et al., 2014*) and triglyceride (*Chen et al., 2008*) concentrations in cafeteria fed dams. The above-cited studies have some differences in the duration and timing of intervention. Therefore, it becomes very difficult to reach a clear conclusion. In the present study, the concentrations of these plasma metabolites were determined at the end of lactation. Therefore, it is crucial to assess the influence of both cafeteria diet and taurine supplementation on these metabolic parameters during pregnancy in future studies.

Previous reports have shown that plasma leptin concentrations of cafeteria diet-fed dams increased in proportion to body fat mass (*Chen et al., 2008*; *Bouanane et al., 2009*; *Jacobs et al., 2014*). While CAF dams displayed higher plasma leptin concentrations in comparison to control dams, CAFT dams did not display hyperleptinemia in the current study. Similarly, in one study increased plasma leptin levels were observed in maternal obesogenic diet group but not in taurine supplemented group although there was no significant effect of taurine (*Li et al., 2013*). *Kim et al. (2012)* reported that leptin levels were significantly lower in taurine supplemented group despite similar body weight and epididymal fat mass to control groups. They suggested that additional studies are needed to elucidate the possible effect of taurine on leptin signaling in adipose tissue (*Kim et al., 2012*). This result is consistent with another report which showed that long term

taurine supplementation did not reduce fat tissue but decreased mRNA expression levels of leptin in white adipose tissue (*Kim et al., 2019*). Decreased circulating adiponectin levels have been demonstrated in high fat and cafeteria diet induced murine models of obesity (*Chaolu et al., 2011*; *Suárez-García et al., 2017*). Interestingly, plasma adiponectin concentrations of both CAF and CAFT groups were markedly greater than CON and CONT groups in the present study. Some studies have shown that body weight reduction resulted in increased adiponectin levels in obesity (*Yang et al., 2001*; *Esposito et al., 2003*). Therefore, weight loss of CAF and CAFT groups during lactation may have lead to higher adiponectin levels in comparison to CON and CONT groups.

Obesity is an independent risk factor for lipid peroxidation and malondialdehyde is one of the most frequently used indicators of lipid peroxidation (*Yesilbursa et al., 2005*; *Marseglia et al., 2014*). High fat and cafeteria diet induced obesity in rats led to elevated malondialdehyde levels in liver in different studies (*Noeman, Hamooda & Baalash, 2011*; *Abd Elwahab et al., 2017*). Similarly, the present study demonstrated that CAF dams had increased plasma malondialdehyde concentrations. Plasma malondialdehyde levels of CAFT dams did not differ from CON and CONT dams, which indicated a partial normalization of the malondialdehyde levels in response to taurine supplementation. It was reported that taurine administration mitigated hepatic oxidative stress through reduction of malondialdehyde levels in the liver of cafeteria fed rats (*Abd Elwahab et al., 2017*). Also, *Ogasawara et al. (1993)* reported that taurine inhibited the production of oxidized low density lipoprotein by reacting with malondialdehyde.

Lower energy contribution of protein in the cafeteria diet may induce nitrogen sparing mechanisms including higher intestinal absorbtion of amino acids and excretion of less urinary and fecal nitrogen (*Esteve et al., 1993*; *Oliva et al., 2017*). In addition, oxidation of amino acids and urea excretion can decrease (*Sabater et al., 2014*; *Oliva et al., 2017*). These mechanisms led to amino acid imbalance and alterations of amino acid concentrations (*Lladó et al., 1995*; *Sabater et al., 2014*). In the current study, both CAF and CAFT dams displayed higher levels of serine and lower levels of tyrosine compared with control dams. Previous studies reported the effects of cafeteria diet on plasma amino acid concentrations with contradictory results (*Salvadó, Segués & Arola, 1991*; *Lladó et al., 1995*; *Pomar et al., 2019*). Offspring suckled by cafeteria diet fed dams exhibited higher circulating levels of serine due to increased hepatic gluconeogenesis (*Pomar et al., 2019*). However, *Salvadó, Segués & Arola (1991)* demonstrated that serine concentrations was lower in the pups exposed to maternal cafeteria diet during lactation than in the control pups. The low serine concentrations of the pups exposed to maternal cafeteria diet were related to contribution of this amino acid to glucose synthesis in the suckling offspring. Also, serine was associated with increased growth rate in the pups exposed to maternal cafeteria diet. The amino acid imbalance observed in this study could be related with a possible maternal trade-off to improve growth and development during suckling period.

In the current study no difference was observed in litter size and sex ratio between groups. This is consistent with previous reports (*Bayol, Simbi & Stickland, 2005*; *Li et al., 2013*; *Li et al., 2015*; *Vithayathil et al., 2018*). Many studies reported conflicting results

about the birth weights of pups exposed to maternal cafeteria diets. Some have found no differences in birth weights (*Bayol, Simbi & Stickland, 2005*; *Chen et al., 2008*; *Speight et al., 2017*; *Kalem et al., 2018*) whereas others have reported higher birth weights (*Bayol, Farrington & Stickland, 2007*; *Sánchez-Blanco et al., 2016*; *Vithayathil et al., 2018*; *Cardenas-Perez et al., 2018*). Also, a meta regression analysis of animal models investigating the effect of maternal obesogenic diet exposure on birthweight demonstrated that this exposure had no effect on birthweight (*Ribaroff et al., 2017*). *Akyol, Langley-Evans & McMullen (2009)* reported that exposure to maternal cafeteria diet led to fetal growth restriction, but they showed increased birth weights in their further study (*Akyol, McMullen & Langley-Evans, 2012*). These differences might occur due to using different food items in cafeteria diet and duration and time of exposure.

Similar to the results of this study, it was reported that maternal taurine supplementation had no effect on birth weights (*Li et al., 2013*). In addition to these outcomes, rise in neonatal mortality was observed in CONT offspring. Similarly, other reports showed that taurine supplementation in the setting of normal pregnancy resulted in increased neonatal mortality (*Li et al., 2013*; *Li et al., 2015*). There are limited data on possible unfavorable effects of taurine in normal pregnancies and underlying mechanisms have not been elucidated, clearly. *Boujendar et al. (2002)* reported that taurine provided in dams fed a control diet induced fetal hypoglycaemia and decreased pancreatic and postnatal body weights. Although in vivo taurine supplementation exerted protective effects on pancreatic islets of the offspring from low protein diet fed dams against cytokine toxicity, islet sensitivity of control animals increased and pancreatic development impaired (*Boujendar et al., 2002*; *Merezak et al., 2004*). These results demonstrated that the effects of taurine supplementation on pregnancy outcomes were closely associated with maternal nutritional background. Future studies should investigate the possible toxicity of taurine supplementation in control pregnancy outcomes. In fact, one limitation of this study could be giving taurine supplementation during pre-gestation since this design complicates translating the current approach and outcomes to human pregnancy. Instead, taurine supplementation could have been administered after mating but this procedure might have masked the potential regressive influence of taurine on obesity development. Therefore, in addition to possible toxicity of taurine supplementation in control pregnancy outcomes, a further investigation can examine addition of taurine supplement after conception in a similar setting.

Exposure to maternal high fat diet during gestation may alter the development of various organs and affect several organ systems. These effects differ by the animal model, timing and duration of the high fat diet exposure as well as the offspring's gender (*Williams et al., 2014*). Previously it has been reported that offspring exposed to maternal obesogenic diet exhibited increased relative heart weight at weaning (*Blackmore et al., 2014*). This study demonstrated that taurine supplementation to cafeteria diet resulted in decreased heart weight in both male and female CAFT offspring at weaning. It was shown that taurine supplementation reduced heart weight in hypertensive rats and was associated with decreased cardiac hypertrophy by displaying antioxidant activity (*Chahine et al., 2010*). In addition, treatment of diabetic animals with taurine reduced heart weight and

this reduction was associated with diminshed cell damage in the diabetic heart (*Tappia et al., 2011*).

## CONCLUSIONS

In conclusion, present data suggest that maternal cafeteria diet led to increased adiposity and malondialdehyde levels, hyperleptinemia and decreased plasma taurine levels. Maternal taurine supplementation did not prevent adiposity but partially normalized cafeteria-induced maternal metabolic dysfunction. The reason why taurine did not have profound protective effects on these metabolic disturbances can be attributable to the amount of taurine consumed by rats. Furthermore, it was showed that taurine supplementation resulted in increased neonatal mortality in control pregnancies, which might be associated with maternal nutritional background. While a few other studies investigated the effects of maternal taurine supplementation on metabolic disturbances induced by high fructose and high fat obesogenic diets (*Li et al., 2013*; *Li et al., 2015*), this study reported the influences of taurine supplementation in a model of maternal cafeteria diet for the first time. The long-term effects of maternal taurine supplementation on offspring and adverse maternal effects in normal pregnancies must be further investigated.

### Funding

This work was funded by the the Scientific And Technological Research Council Of Turkey (No.115S538) and Scientific Research Projects Coordination Unit of Hacettepe University (No.TDK-2019-17813). The funders had no role in study design, data collection and analysis, decision to publish, or preparation of the manuscript.

### Grant Disclosures

The following grant information was disclosed by the authors:
Scientific And Technological Research Council Of Turkey: 115S538.
Hacettepe University: TDK-2019-17813.

### Competing Interests

The authors declare that they have no competing interests.

### Author Contributions

- Arzu Kabasakal Çetin performed the experiments, analyzed the data, prepared figures and/or tables, authored or reviewed drafts of the paper, and approved the final draft.
- Tuğba Alkan Tuğ performed the experiments, prepared figures and/or tables, and approved the final draft.
- Atila Güleç performed the experiments, prepared figures and/or tables, and approved the final draft.
- Aslı Akyol conceived and designed the experiments, analyzed the data, prepared figures and/or tables, authored or reviewed drafts of the paper, and approved the final draft.

## Animal Ethics

The following information was supplied relating to ethical approvals (i.e., approving body and any reference numbers):

Ethics Committee of Hacettepe University provided full approval for this research (2015/01).

## Data Availability

The raw measurements are available in the Supplemental Files.

## Supplemental Information

Supplemental information for this article can be found online at http://dx.doi.org/10.7717/peerj.11547#supplemental-information.

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
