# Peer review of "Effects of maternal taurine supplementation on maternal dietary intake, plasma metabolites and fetal growth and development in cafeteria diet fed rats"

_PeerJ, doi:10.7717/peerj.11547_

## Round 0.1 · original submission · Major Revisions

Dear Authors:
Please heed all of the reviewers' comments when making a new version of the manuscript, if you think you can answer all of them, especially those of reviewer #2.

Reviewer 1 ·

Basic reporting

The work is well presented, background and hypothesis clear .
Figures and tables included enough information to understand the work

Experimental design

Just to know, how the taurine concentration was selected?

Validity of the findings

It is a relevant study, there is not specific information of that

Additional comments

The aim of this study was to analyze the protector effect of taurine on cafeteria dietary intake in rats during gestation and lactation.
The authors showed interesting results supporting partial normalization of cafeteria-induced maternal metabolic dysfunction by taurine; however taurine supplementation did not prevent adiposity. Even more, taurine supplementation cause increase in neonatal mortality in the control group.
My main concern is to the model; it is expected obesity induction by the cafeteria diet (CAF). I think it is difficult to know the energy and nutrients intake, since cafeteria diet vary every day. How that was calculated? How the taurine concentration was selected?
Although some metabolic parameters indicate increase in fat tissue, no differences in body weight gain were observed compared to the control in gestational period. In fact, the fat intake was increased and protein and carbohydrates decreased.
As Authors mentioned, taurine concentration could be low; in fact the intake of water ( containing taurine ) was considerable reduced in the CAF groups.
Probably some results are underestimated; although the statistical significance is clear, the standard error in several cases is high.

Reviewer 2 ·

Basic reporting

Data and Figures/Tables:

Please can authors confirm that all data were tested for normality and the parametric statistical tests applied are therefore appropriate.

In methods it is stated that litters were culled to a max of 8 (4M/4F if possible). In results it states that litter sizes were comparable and male:female ratios unaffected so one would expect similar n’s across all groups in Figure 4 (offspring body weight). But n’s range from n=12-24. Can authors provide an explanation for this?

In methods it is also stated that at the end of lactation, mothers and three male and three female offspring from each litter were culled. Table 5 n numbers are inconsistent with this: there should be n=18 for male and female pups from the control group (based on n=6 control dams) and n=21 for the remaining experimental groups (based on n=7 dams). Furthermore, control females are n=24 which would mean 4 were culled per litter rather than 3. Can authors provide an explanation for this?

There are inconsistencies in reporting of data: Table 1 data is intake in g/day, therefore not normalised to body weight. However authors report taurine consumed in mg/g body weight/day (starting on line 214), which is normalised to body weight. Do the effects seen in Table 1 remain when body weight is entered into analyses as a covariate? Conversely, how are the estimated values for taurine consumption affected if not normalised to body weight? Authors should consider if it is appropriate to normalise these data to body weight.

Table 2 – organ weight is expressed as % of body weight. This seems inappropriate when there have been fluctuations in maternal body weight over the course of the study (e.g. CAFT dams were heavier pre-gestation, comparable during pregnancy, and then lighter during lactation). It is recommended these data are not expressed relative to body weight (as was done in their 2009 study).

On Figures 2 and 3 it would be helpful to indicate pre-gestation, gestation, and lactation.

Authors should check all figure legends explain what is meant by letters a/b/c (this information appears to be missing from figures 2, 3, and 4.

Experimental design

The research question and its relevance:

The relevance of the study in terms of its translatability is questionable. What is the rationale for giving taurine pre-pregnancy? Translating this in to human pregnancy would require obese women to take taurine supplements prior to conceiving, for which there would be inevitable issues with compliance. Women are more likely to take supplements once pregnant, therefore a more appropriate study would involve administering the taurine after mating. This point should be raised by authors in Discussion as a limitation of the study. Related to this, authors state that they aimed to examine whether taurine given with cafeteria diet during gestation and lactation exerts any protective effects. But, as mentioned, rats were given the diet +/- taurine for 8 weeks prior to mating. Therefore line 111 should be amended so that it reads ‘…examine whether taurine given with cafeteria diet prior to gestation, during gestation and lactation exerts any protective effects…’
In terms of protective effects, it is not clear what the authors expect taurine to protect against. In the abstract it is written: Maternal obesity may disrupt the developmental process of the fetus during gestation in rats. This study aimed to examine the effect of maternal cafeteria diet and/or taurine supplementation on maternal dietary intake (what is the rationale for this? Does taurine supplementation have the potential to affect appetite?), plasma metabolites, fetal growth and development. However, the design of the study has prevented a thorough assessment of fetal growth and development because most assessments were made at weaning rather than at the end of gestation. The cafeteria diet used in the current study is the same as first described by the same authors (Akyol et al) in 2009. In the 2009 study, fetal growth restriction was observed in the CAF-CAF diet. However, this phenotype was not reproduced in the current study. Authors fail to highlight in Discussion this discrepancy between their original study which observed growth restriction and their current study which did not observe any effect on birthweights. Authors should draw comparisons where possible between the current findings and findings from the Akyol et al 2009 study in Discussion.

The aims of the study as outlined in the abstract and in Discussion (line 282) suggest that the focus is on fetal/offspring health, but a large proportion of the study relates to maternal metabolic health measured when pups were at weaning. It is unclear why this timepoint was chosen to assess maternal metabolic health. Can authors comment on the rationale and relevance to human pregnancy?

Validity of the findings

Authors refer to the cafeteria diet as a robust model of inducing dietary obesity in laboratory animals. But it is not confirmed if there was maternal obesity at the time of mating as a result of the cafeteria diet in their current study. CONT appear to be the heaviest at the time of mating (lines 224-225). Please can authors include a sentence that explicitly states if the cafeteria diet had led to obesity by the time of mating, thereby confirming this was a true model of maternal obesity.

How neonatal morbidity was calculated is unclear. Was it an average of litters? Was it relative to born litter size or relative to litter size after culling? Was it comparable across litters and between sexes, or was it skewed in any way? Can authors comment on how soon after birth the pups died? Did some die after culling to a maximum of 8 pups per litter? The answers to these questions are important for determining the validity of these data.

Additional comments

Minor comments:

Intro, lines 83-85. The sentence ‘The identification of obesity related complications during pregnancy and developing effective interventions as early as possible to prevent the development of obesity is vital’ does not make sense. How can an intervention prevent the development of obesity in an individual who is already obese? Do the authors mean to say development of complications?

Lines 109-100 state 'To date none of the studies have investigated the possible protective role of taurine within a maternal cafeteria diet model on pregnancy outcomes in rats'. However the Li et al 2013 study has, therefore this sentence should be removed or modified.

Authors should speculate in Discussion on how taurine supplementation could have exerted the positive effects (no maternal hyperleptinemia or increase in MDA) in the obese model if plasma taurine was not increased. Did authors make a direct comparison of plasma taurine between CAF and CAFT? The data suggest this is likely to show significantly higher taurine levels in CAFT.

---

## Round 0.2 · accepted · Accept

Please make the changes that reviewer#2 asks, for greater consistency.

Reviewer 1 ·

Basic reporting

no comment

Experimental design

no comment

Validity of the findings

no comment

Additional comments

Authors answer my questions and did appropiate corrections in the manuscript.

It is clear that new experiments should be carried out to understand the possible role of taurine on obesity

Reviewer 2 ·

Basic reporting

Authors have thoroughly addressed all comments raised in my original review of this manuscript with the exception of the following minor point:
- Pre-gestation, gestation and lactation indicators need adding to Figure 3, as has been done for Figure 2.

Table 2 data is now presented as raw organ weight rather than as a % of body weight. For consistency, it is suggested that the data in Table 5 (offspring organ weight) is also presented as raw weight instead of as a % of body weight.

Experimental design

No comment.

Validity of the findings

No comment.

Additional comments

Thank you to the authors for thoroughly addressing all comments raised in my original review of this manuscript.